# Re-Evaluating the Protective Effect of Hemodialysis Catheter Locking Solutions in Hemodialysis Patients

**DOI:** 10.3390/jcm8030412

**Published:** 2019-03-25

**Authors:** Chang-Hua Chen, Yu-Min Chen, Yu Yang, Yu-Jun Chang, Li-Jhen Lin, Hua-Cheng Yen

**Affiliations:** 1Division of Infectious Disease, Department of Internal Medicine, Changhua Christian Hospital, Changhua 500, Taiwan; 2Center for Infection Prevention and Control, Changhua Christian Hospital, Changhua 500, Taiwan; 3344@cch.org.tw; 3Program in Translational Medicine, National Chung Hsing University, Taichung County 402, Taiwan; 4Rong Hsing Research Center for Translational Medicine, National Chung Hsing University, Taichung County 402, Taiwan; 5Department of Pharmacy, Changhua Christian Hospital, Changhua 500, Taiwan; 30855@cch.org.tw; 6Division of Nephrology, Department of Internal Medicine, Changhua Christian Hospital, Changhua 500, Taiwan; 2219@cch.org.tw; 7Epidemiology and Biostatistics Center, Changhua Christian Hospital, Changhua 500, Taiwan; 83686@cch.org.tw; 8Department of Neurosurgery, Changhua Christian Hospital, Changhua 500, Taiwan; 90211@cch.org.tw

**Keywords:** effect, protection, catheter, hemodialysis, meta-analysis, trial sequential analysis

## Abstract

Catheter-related bloodstream infections (CRBSIs) and exit-site infections (ESIs) are common complications associated with the use of central venous catheters for hemodialysis. The aim of this study was to analyze the impact of routine locking solutions on the incidence of CRBSI and ESI, in preserving catheter function, and on the rate of all-cause mortality in patients undergoing hemodialysis. We selected publications (from inception until July 2018) with studies comparing locking solutions for hemodialysis catheters used in patients undergoing hemodialysis. A total of 21 eligible studies were included, with a total of 4832 patients and 318,769 days of catheter use. The incidence of CRBSI and ESI was significantly lower in the treated group (citrate-based regimen) than in the controls (heparin-based regimen). No significant difference in preserving catheter function and all-cause mortality was found between the two groups. Our findings demonstrated that routine locking solutions for hemodialysis catheters effectively reduce the incidence of CRBSIs and ESIs, but our findings failed to show a benefit for preserving catheter function and mortality rates. Therefore, further studies are urgently needed to conclusively evaluate the impact of routine locking solutions on preserving catheter function and improving the rates of all-cause mortality.

## 1. Introduction

### 1.1. Variety of New Strategies for Locking Solutions to Avoid Catheter Infection and Catheter Malfunction in Hemodialysis Patients

Infections are widely prevalent in patients on chronic hemodialysis, and mortality from infection account for 10% of deaths observed in patients undergoing hemodialysis [1]. The use of central venous catheters in hemodialysis has been associated with catheter-related bloodstream infections (CRBSIs) and exit-site infections (ESIs) [2,3,4]. Although recent efforts have minimized the use of catheters, the proportion patients with end-stage renal disease undergoing dialysis using central venous catheters has not yet declined [5]. Protective strategies against CRBSI and catheter malfunction are necessary [6], and to this end, the use of heparin as a routine locking solution for central venous catheters has become an accepted clinical practice [4]. However, heparinized locking solutions might cause unintended complications, such as systemic anticoagulation effects, bleeding episodes, heparin-induced thrombocytopenia, and susceptibility to bacterial biofilm formation [7,8,9]. A variety of new locking solutions have been developed; this includes citrate, which has antimicrobial properties [10,11,12]. However, the disadvantages of citrate compared with heparin have been raised and included the ability of avoiding catheter malfunction, citrate toxicity, and induction of cardiac arrhythmia [13]. Weijmer et al. showed that a 30% citrate solution was superior to heparin in preventing CRBSI [14]. In contrast, other studies have reported that the use of citrate does not have an advantage over heparin in preventing CRBSI [4,12]. Currently, the findings of the studies comparing citrate with heparin locking solutions are inconclusive for protecting against CRBSI and ESI and preserving the catheter function. Clinicians question if locking solutions should be considered a modifiable risk factor for CRBSIs in patients undergoing hemodialysis. Furthermore, the recommended locking solution for the routine care of patients undergoing hemodialysis continue to remain questionable.

### 1.2. Rationale for Re-Evaluating the Protective Effect of Hemodialysis Catheter Locking Solutions in Hemodialysis Patients

Routine locking solutions for hemodialysis catheters are recommended with category II evidence according to the guideline by the Healthcare Infection Control Practices Advisory Committee in 2011 [15]; however, there are some limitations of the studies providing the current and update evidence. Mostly, conclusions of meta-analysis could be influenced by the heterogeneity between individual studies and insufficient information size. Quantification of the required information size [16] is important to ensure the reliability of the data. In addition, current meta-analyses lack information size calculation [17,18,19,20,21,22,23,24]. Additionally, the incidence of CRBSI is difficult to evaluate because of their subjectivity for case finding, lack of specificity, and high inter-observer variability. CRBSI is associated with high morbidity and mortality in patients undergoing hemodialysis [1], and the prevention of CRBSI and ESI is becoming increasingly essential. Given these limitations, we performed a meta-analysis and trial sequential analysis to assess the impact of routine locking solutions on the incidence of CRBSI and ESI, in preserving catheter function, and on the rate of all-cause mortality in patients undergoing hemodialysis. We grouped the eligible publications according to combination regimen, antimicrobial activity, and concentration of the locking solutions; thereafter, we grouped according to the study design to assess its potential effect on the reported outcomes.

## 2. Experimental Section

### 2.1. Search Strategy and Inclusion Criteria

The study was conducted in accordance with the Declaration of Helsinki, and the protocol was approved by the institutional review board of Changhua Christian Hospital (CCH IRB No. 180801). From the earliest record to July 2018, we searched PubMed, Scopus, Cochrane Central Register of Controlled Trials, Cochrane Database of Systematic Reviews, ClinicalTrials.gov, Embase, and Web of Science databases for studies on locking solutions for central venous catheters used in hemodialysis of patients. Full search strategies for each database are available in the Appendix B. The reference lists of the eligible publications were manually reviewed for relevant studies. Articles published in languages other than English or those with no available full text were excluded.

We included all trials and studies that provided data on one or more of our target outcomes for both the treated group and control group: CRBSIs and ESIs. Two investigators (CHC and YMC) independently reviewed potential trials and studies for inclusion. Disagreements were resolved by consensus. We also tried to contact the corresponding authors of selected papers to provide clarifications and missing data where needed.

### 2.2. Definition of Study Outcomes

Based on the original studies, the treated group comprised of patients undergoing hemodialysis using citrate as the locking solution for central venous catheters; for the control group heparin was used as the locking solution (Table 1). The outcomes of the original studies were included in this meta-analysis. The primary outcomes included (1) CRBSI, defined as bacteremia caused by an intravenous catheter, and (2) ESI, defined as the development of a purulent redness around the exit site that did not result from residual stitches. The secondary outcomes included (1), the need to remove the catheter due to catheter malfunction; and (2) the need for thrombolytic treatments due to catheter malfunction; and (3) all-cause mortality at any timeframe. Incidence was presented as the number of episodes per catheter or per patient depending on the available data.

### 2.3. Data Extraction and Quality Assessment

Two reviewers examined all retrieved articles and extracted data using a pre-determined form, recording the name of the first author, year of publication, country where the study was conducted, study design (RCT or observational studies), demographic and disease characteristics of participants, number of participants enrolled, and quality assessment of each study. Each reviewer independently evaluated the quality of the eligible studies, using Jadad scoring [25] for the RCTs and the Newcastle-Ottawa quality assessment scale [26] for the comparative experimental studies.

### 2.4. Data Synthesis and Analysis

The outcomes were measured by determining the odds ratios (ORs). A random effects model was used to pool individual ORs. Analyses were performed with the Comprehensive Meta-Analysis software version 3.0 (Biostat, Englewood, NJ, USA). Between-trial heterogeneity was determined using *I*^2^ tests; values > 50% were regarded as considerable heterogeneity [27]. Funnel plots and Egger’s test were used to examine potential publication bias [27]. Statistical significance was defined as *p* < 0.05, except for the determination of publication bias where *p* < 0.10 was considered significant. This study was conducted and reported in accordance with the Preferred Reporting Items for Systematic Reviews and Meta-Analyses (PRISMA) statement (Appendix A) [28].

In trial sequential analyses, the inconsistence of heterogeneity (*I*^2^) adjusted by determining the required information size. The required information size was calculated with an intervention effect of a 10% relative risk reduction, an overall 5% risk of a type I error, and a 20% risk of a type II error. All trial sequential analyses were performed using TSA version 0.9 Beta (www.ctu.dk/tsa/, Copenhagen Trial Unit, Copenhagen, Denmark).

## 3. Results

### 3.1. Eligible Studies

The literature search yielded 458 potentially eligible articles. By screening the abstracts, we removed 350 irrelevant articles. The remaining 100 articles were assessed further by full-text reading, of which 79 were excluded (Figure 1). Thus, 21 selected articles comparing citrate with heparin lockings for central venous catheters used in hemodialysis were included in this meta-analysis [4,6,8,12,13,14,17,18,19,20,21,22,23,24,29,30,31,32,33,34,35].

The studies published in the selected articles were conducted from the earliest record to July 2018, with a total of 4832 patients and 318,769 total days of catheter use. Six studies compared citrate alone with heparin lockings; 14 studies tested regimens of citrate and other antimicrobials (gentamicin, taurolidine, methylparaben, methylene blue, and propylparaben) with heparin lockings; and two studies compared ethanol or combination solution (citrate, heparin and taurolidine) with non-heparin locking. Studies were conducted in North America (5 studies), South America (1), Europe (12), and Asia (3). A variety of end points were used in these studies. Most studies reported on CRBSI (17 studies [4,6,8,12,13,14,17,19,21,22,23,24,30,31,32,33,34]), followed by ESI (11 studies [4,8,12,14,17,18,19,24,30,31,33]), catheter removal for poor flow (9 studies [6,8,12,14,18,24,29,31,33]), thrombolytic treatment (8 studies [4,8,14,17,18,32,33]), and mortality (5 studies [6,14,19,32,33]). The characteristics of the studies fulfilling the inclusion criteria are listed in Table 1. Thirteen studies were identified as RCT, and 6 studies were not double-blinded (Table 1).

### 3.2. Pooled Odds for Primary Outcomes and Subgroup Analysis

#### 3.2.1. Catheter-Related Bloodstream Infection (CRBSI)

Seventeen studies (1731 patients; 217,128 catheter days) reported on CRBSI. The incidence of CRBSI was significantly lower in the treated group compared with the control group (OR, 0.424; 95% CI, 0.267–0.673; *p* < 0.001) (Figure 2). CRBSI subgroup analysis showed that the OR appeared to have a tendency to favor the treatment groups with either the combined regimen (OR, 0.206; 95% CI, 0.058–0.730; *p* = 0.027), the single regimen (OR, 0.289; 95% CI, 0.083–0.365; *p* = 0.037), a regimen containing antibiotics (OR, 0.136; 95% CI, 0.051–0.365; *p* = 0.002), or a low concentration of a major regimen (OR, 0.421; 95% CI, 0.186–0.956; *p* = 0.039; Table 2).

#### 3.2.2. Exit-Site Infection (ESI)

A total of 11 RCTs (2,425 patients; 231,086 catheter days) described ESI. The incidence of ESI was significantly lower in the treated group compared with the control group (OR, 0.627; 95% CI, 0.441–0.893; *p* = 0.001; Figure 3). Further focusing at exit-site infection (Table 3), the subgroup analysis (combined regimen, regimen containing antibiotic, and concentration of regimen for exit-site infection) disclosed no significant differences between any groups except for combined regimen.

### 3.3. Pooled Odds for Secondary Outcomes and Subgroup Analysis

#### 3.3.1. Catheter Withdrawal Due to Malfunction

Nine studies (1826 patients; 205,163 catheter days) reported catheters being removed for poor blood flow. As shown in Figure 4, no difference was identified between the two groups (OR, 0.696; 95% CI, 0.397–1.223; *p* = 0.208). Further subgroup analysis (combined regimen, regimen containing antibiotic, and concentration of regimen for catheter removal due to catheter malfunction) failed to reveal any differences between any groups (Table 4).

#### 3.3.2. Thrombolytic Treatment Due to Catheter Malfunction

Overall, in eight RCTs (2092 patients; 220,460 catheter days) included in this meta-analysis the patients underwent thrombolytic treatment [4,8,14,17,18,32,33]. The incidence of thrombolytic treatment was not significantly lower in the treated group compared with the control group using the random-effects model (OR, 1.105; 95% CI, 0.655–1.573; *p* = 0.946; Figure 5). Thrombolytic treatment subgroup analysis showed no differences in the OR between the two groups (Table 5).

#### 3.3.3. All-Cause Mortality

The meta-analysis included five RCTs (2,327 patients) comparing all-cause mortality rate between the two groups; no significant difference was identified (OR, 0.909; 95% CI, 0.580–1.423; *p* = 0.676; Figure 6). The corresponding subgroup analysis (combined regimen, regimen containing antibiotic, and concentration of regimen for all-cause mortality) showed no apparent differences between the two groups (Table 6).

### 3.4. Pooled Odds for Outcomes in Trial Sequential Analysis

In trial sequential analysis between the treated and control groups, the overall OR of CRBSI was 0.439 (95% CI, 0.290–0.668; *p* < 0.001; Figure 7a), the OR of ESI was 0.644 (95% CI, 0.469–0.883; *p* = 0.006; Figure 7b), the OR of the need to remove the catheter for catheter malfunction was 0.746 (95% CI, 0.431–1.293; *p* = 0.151; Figure 7c), the OR of the need to receive thrombolytic treatment for catheter malfunction was 1.015 (95% CI, 0.655–1.573; *p* = 0.461; Figure 7d), and the OR of all-cause mortality was 0.976 (95% CI, 0.663–1.439; *p* = 0.296; Figure 7e).

### 3.5. Funnel Plot for the Overall OR of the Included Studies among Four Outcomes

We examined possible sources of underlying heterogeneity across studies. With regards to OR heterogeneity, the *I*^2^ value was calculated in both the overall studies included. In the funnel plot of the OR for evaluation event, the *I*^2^ value of CRBSI was 70.1% (*p* = 0.303, Figure 8a), ESI was 28.0% (*p* = 0.010; Figure 8b), the need to remove the catheter for catheter malfunction was 55.9% (*p* = 0.208; Figure 8c), the need to receive thrombolytic treatment for catheter malfunction was 88.69% (*p* = 0.946; Figure 8d), and all-cause mortality was 88.6% (*p* = 0.804; Figure 8e). 

## 4. Discussion

Our meta-analysis and trial sequential analysis shows that routine locking solutions for hemodialysis catheters could effectively reduce the incidence of CRBSI and ESI. Our current meta-analysis, based on 21 selected studies with a total of 6118 participants, showed that the incidence of CRBSI and ESI significantly decreased in the treated group relative to the control group, that is less infections when using citrate or citrate mixtures versus heparin. Moreover, we found no significant difference in preserving catheter function, including in the need for catheter withdrawal or for thrombolytic treatment due to catheter malfunction, between the treated and control groups. We found no significant alteration in all-cause mortality between the two groups. The lack of statistical significance may not only be due to the heterogeneity and underlying variance in the outcomes of each regimen, but also due to inadequate required information sizes, as revealed by the trial sequential analysis. Regular locking care with citrate is standard practice for patients undergoing hemodialysis in many healthcare institutes, but not in some countries including Taiwan. Our updated review suggests that the role of routine locking solutions in preventing CRBSI and ESI in hemodialysis patients is robust. However, it does not show a benefit in preserving catheter function in hemodialysis patients, including in the need to remove catheters or in the need for thrombolytic treatment for catheter malfunction.

The current study shows that the incidence of CRBSI significantly decreased in the treated group relative to the control group, which is consistent with previous studies [36,37]. Subgroup analyses based on the type of locking solutions for hemodialysis catheters revealed that the usage of citrate-base regimens was associated with a lower incidence of CRBSI [4,14]. Our subgroup analysis for the concentration of citrate used showed that the incidence of CRBSI was similar in treated group, although the American Society of Diagnostic and Interventional Nephrology and the European Renal Best Practice recommend 4% citrate to be used as a catheter locking solution [38,39]. In some countries, including Taiwan, 4% citrate is still not routinely used in locking solutions for hemodialysis catheters. The current meta-analysis emphasizes that 4% citrate shows a benefit and could be routinely used as a locking solution for hemodialysis catheters.

Our current study shows that the incidence of ESI is significantly decreased in the treated group compared with the control group. Our result is in agreement with previous studies [14,19]. In some studies, patients received additional antibiotic ointments at the exit site during dressing changes, which could reduce the incidence of ESI [8,14,40]. After subgroup categorization, there is no significant difference between two groups except for combined regimen, which could result from the heterogeneity of the included studies and inadequate information size.

We found no significant difference in preserving catheter function between the treated and control groups, including the need to remove catheters or the need for thrombolytic treatment. However, Yahav et al. reported that citrate reduced catheter removals [41]. This incongruity may arise from the following: (1) Variation in enrollment criteria and definitions for the spectrum of catheter removal and (2) the number of cases is still limited because the meta-analysis information size does not meet the required information size. Concerning thrombolytic treatments and thrombosis episodes, our report is similar to previous studies [41,42]. Focusing on the need to remove catheters and to receive thrombolytic treatment for catheter malfunction, further large-scale RCTs are necessary to elucidate this issue for preserving catheter function.

The possible association between the two groups and all-cause mortality was not statistically significant in the current study (OR, 0.909; 95% CI, 0.580–1.423; *p* = 0.676). Subgroup analysis showed no difference in all-cause mortality. Mortality due to CRBSIs or ESIs account around one-tenth of all hemodialysis patient deaths [1,2,3,4]. Protective strategies with locking solutions to prevent CRBSIs and ESIs in hemodialysis patients still cannot decrease the mortality rate. Further large-scale RCTs are necessary to elucidate modifiable risk factor for decreasing morality in hemodialysis patients.

Guidelines for the Prevention of Intravascular Catheter-Related Infections has been published by the Center for Disease Control and Prevention [15], which recommends using prophylactic antimicrobial locking solution in patients undergoing hemodialysis who have a history of multiple CRBSI, despite optimal maximal adherence to aseptic techniques (Category II). This recommendation has been embraced by some dialysis centers due to the low execution rate of locking solutions in preventing CRBSI in hemodialysis patients. In fact, many challenges persist in managing daily care in dialysis centers, such as a lack of safety locking solutions for hemodialysis catheters, lack of a designated health-care workers to perform locking care, limited training on catheter care among health-care workers of dialysis centers, potential hemodialysis patients’ noncompliance due to discomfort, as well as health-care workers being unable to maintain high adherence rates in conducting care procedures.

The current study has several limitations. Firstly, the enrolled trials and studies included in the primary analysis dealt with different indications for outcome measures by randomizing a variety of patient groups in different clinical settings. Thus, there is the risk of introducing potentially heterogeneity. Additionally, it is difficult to perform a subgroup analysis based on conditions, such as catheter type, heparin dosage, and other differences in individual unit practices. Secondly, differences in the study individuals, disease severity, setting, and type of infections between individual studies made the study population highly heterogeneous. The *I*^2^ value for OR heterogeneity ranged from 25% to 50%, and this heterogeneity would impact the findings of this meta-analysis. Thus, the influence of measurement precision was considered when reporting treatment effectiveness using ORs. Due to the lack of adjusted data in our selected trials, we compiled the unadjusted ORs. We therefore suggest that future similar trials should record serial changes in catheter function and infection status to provide a more accurate indication of clinical effectiveness. Regardless of aforementioned limitations, we have minimized bias throughout the process by our methods of study identification, data selection, and statistical analysis, as well as in our control of publication bias. These steps should strengthen the stability and accuracy of the meta-analysis. Our findings of this meta-analysis are reliable to provide suggestions for improving clinical care.

## 5. Conclusions

In conclusion, our study demonstrated that routine locking solutions for hemodialysis catheters could effectively reduce the incidence of CRBSI and ESI. Our findings showed no benefit of routine locking solutions for hemodialysis catheters in decreasing all-cause mortality as well as preserving catheter function, including in the need to remove catheters and in the need to receive thrombolytic treatment, both due to catheter malfunction. The latter results lack statistical significance and the comparisons are limited due to the heterogeneity of the included trials and inadequate information size. Therefore, further well-conducted observational studies and randomized controlled trials are urgently needed to conclusively evaluate the impact of routine locking solutions on preserving catheter function and improving the rates of all-cause mortality.

## Figures and Tables

**Figure 1 jcm-08-00412-f001:**
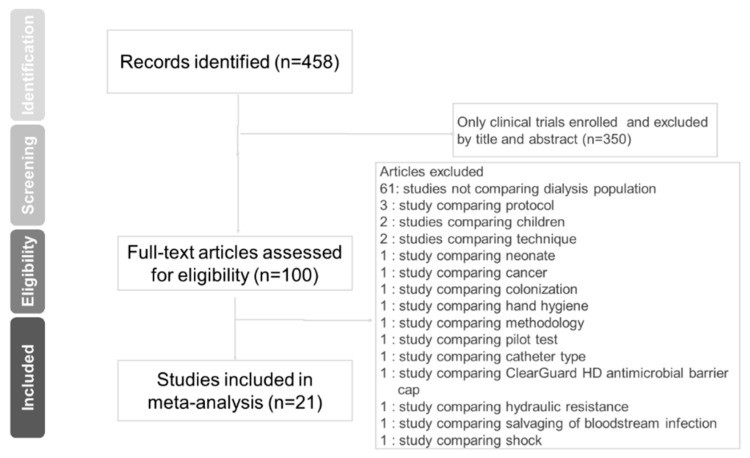
Preferred reporting items for systematic reviews and meta-analyses (PRISMA) flow diagram for the search and identification of the included studies.

**Figure 2 jcm-08-00412-f002:**
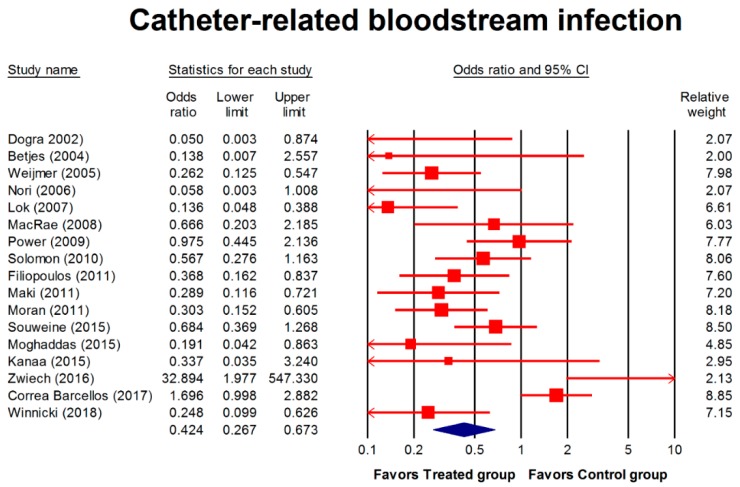
Forest plot of the overall odds ratios for catheter-related bloodstream infection in the treated group versus the control group. The random model of overall odds ratio showed a significant overall effect of interventions in reducing the risk for developing catheter-related bloodstream infections as compared with the control condition (OR, 0.424; 95% CI, 0.267–0.673; *p* < 0.001).

**Figure 3 jcm-08-00412-f003:**
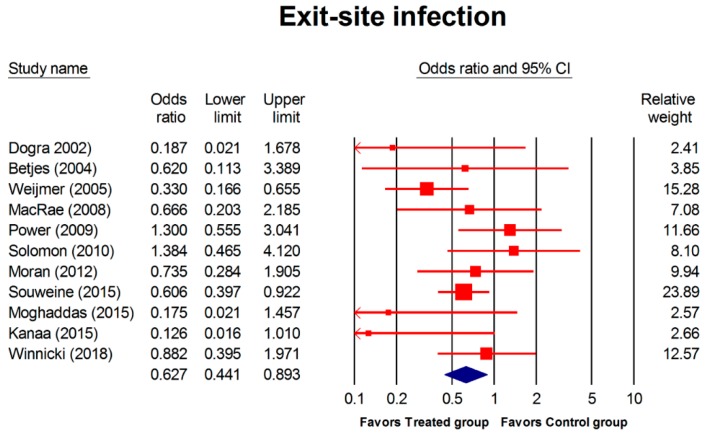
Forest plot of the overall odds ratios for exit-site infection in treated group versus the control group. The random model of overall odds ratio for exit-site infection showed a significant overall effect of interventions in reducing the risk for developing exit-site infection as compared with the control condition (OR, 0.627; 95% CI, 0.441–0.893; *p* = 0.001).

**Figure 4 jcm-08-00412-f004:**
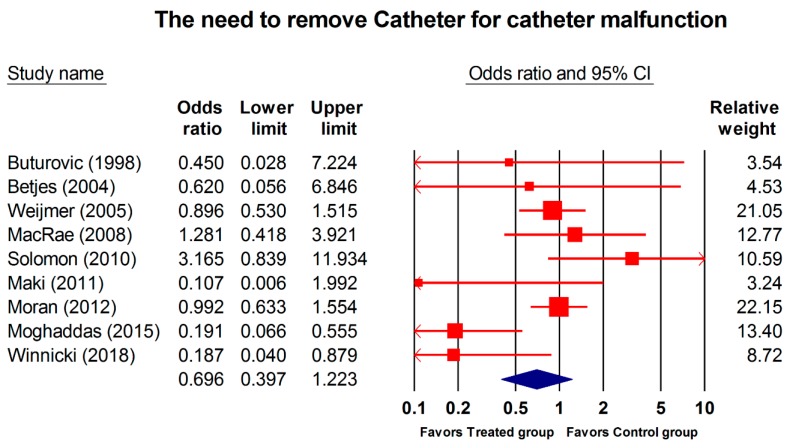
Forest plot of the overall odds ratios for catheter removal due to catheter malfunction in the treated group vs. the control group. The random model of overall odds ratio for the need to remove the catheter for malfunction showed a significant overall effect of the interventions in reducing the risk for catheter removal compared with the control condition (OR, 0.696; 95% CI, 0.397–1.223; *p* = 0.208).

**Figure 5 jcm-08-00412-f005:**
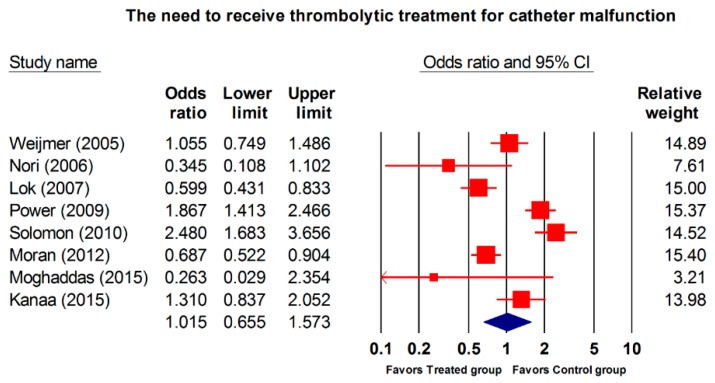
Forest plot of the overall odds ratios for thrombolytic treatments for catheter malfunction in the treated group versus the control group. The random model of overall odds ratio for the need to administer thrombolytic treatment for catheter malfunction showed a significant overall reduced risk for receiving thrombolytic treatments with interventions as compared with the control condition (OR, 1.105; 95% CI, 0.655–1.573; *p* = 0.946).

**Figure 6 jcm-08-00412-f006:**
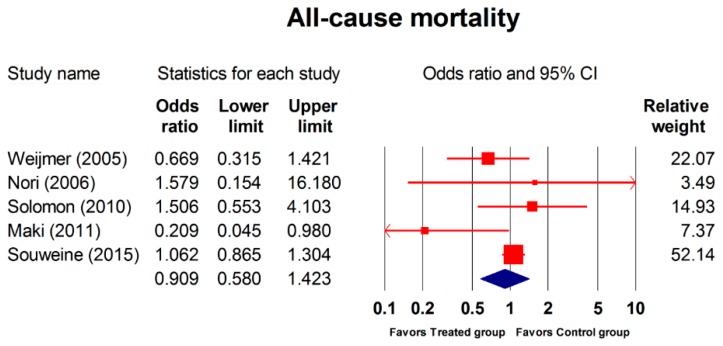
Forest plot of the overall odds ratios for all-cause mortality in the treated group versus the control group. The random model of overall odds ratio for all-cause mortality rate showed a significant overall effect of the interventions in reducing mortality rate as compared with the control condition (OR, 0.909; 95% CI, 0.580–1.423; *p* = 0.676).

**Figure 7 jcm-08-00412-f007:**
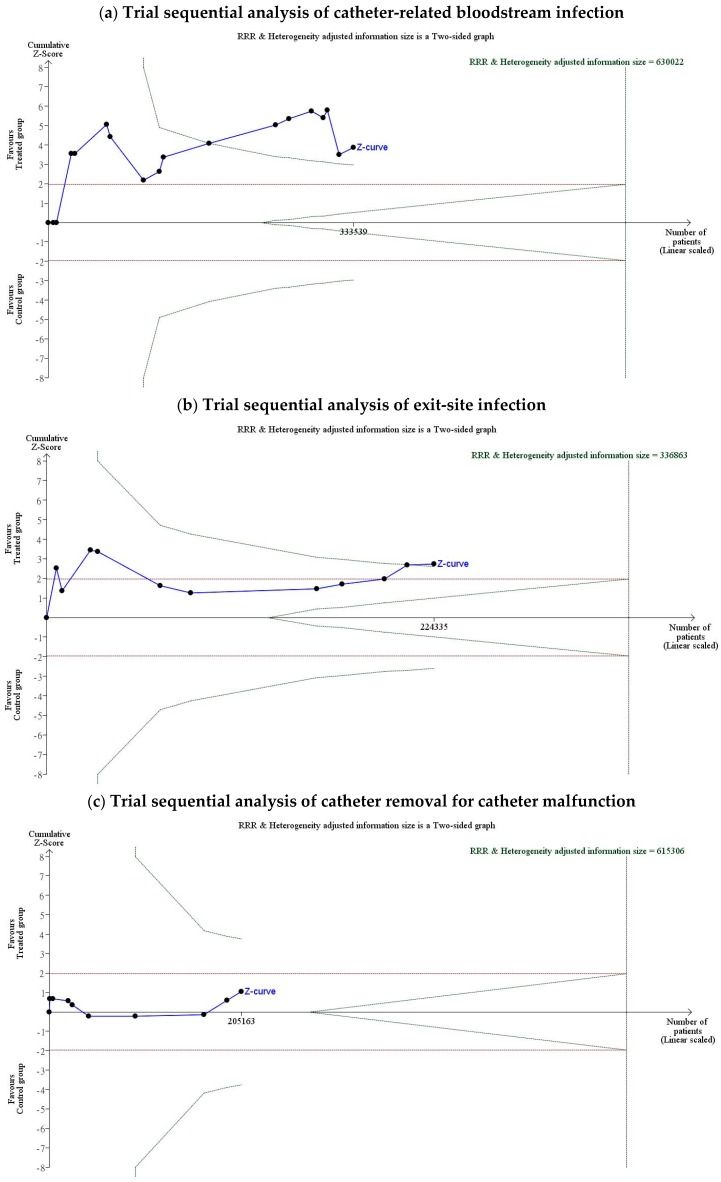
Trial sequential analysis of the odds ratio for evaluation event: (**a**) Trial sequential analysis of catheter-related bloodstream infection. Trial sequential analysis of 17 studies with a lower risk of bias in reporting catheter-related bloodstream infection, with a control event proportion of 17%, diversity of 45%, type I error of 5%, power of 80%, and relative risk reduction of 30%. The required information size of 630,022 was not reached and none of the boundaries for benefit, harm, or futility were crossed, leaving the meta-analysis inconclusive at a 30% relative risk reduction. The overall OR of CRBSI was 0.439 (95% CI, 0.290–1.668; *p* < 0.001); (**b**) trial sequential analysis of exit-site infection. Trial sequential analysis of eleven studies with low risk of bias reporting exit-site infection, with a control event proportion of 17%, diversity of 30%, type I error of 5%, power of 80%, and relative risk reduction of 30%. The required information size of 336,863 was not reached and none of the boundaries for benefit, harm, or futility were crossed, leaving the meta-analysis inconclusive at a 30% relative risk reduction. The OR of ESI was 0.644 (95% CI, 0.469–0.883; *p* = 0.006); (**c**) trial sequential analysis of nine studies with a lower risk of bias reporting the need to remove the catheter for catheter malfunction, with a control event proportion of 17%, diversity of 71%, type I error of 5%, power of 80%, and relative risk reduction of 30%. The required information size of 625,306 were not reached and none of the boundaries for benefit, harm, or futility were crossed, leaving the meta-analysis inconclusive at a 30% relative risk reduction. The OR of the need to remove the catheter for catheter malfunction was 0.746 (95% CI, 0.431–1.293; *p* = 0.151); (**d**) trial sequential analysis of thrombolytic treatments for catheter malfunction. Trial sequential analysis of nine studies with low risk of bias reporting the need to receive thrombolytic treatment for catheter malfunction, with a control event proportion of 17%, diversity of 91%, type I error of 5%, power of 80%, and relative risk reduction of 30%. The required information size of 615,306 were not reached and none of the boundaries for benefit, harm, or futility were crossed, leaving the meta-analysis inconclusive at a 30% relative risk reduction. The OR of the need to receive thrombolytic treatment for catheter malfunction was 1.015 (95% CI, 0.655–1.573; *p* = 0.461); (**e**) trial sequential analysis of all-cause mortality. Trial sequential analysis of five studies with a lower risk of bias reporting all-cause mortality, with a control event proportion of 17%, diversity of 78%, type I error of 5%, power of 80%, and relative risk reduction of 30%. The required information size of 8419were not reached and none of the boundaries for benefit, harm, or futility were crossed, leaving the meta-analysis inconclusive at a 30% relative risk reduction. The OR of all-cause mortality was 0.976 (95% CI, 0.663–1.439; *p* = 0.296). Notes: The solid blue line is the cumulative Z-curve. The vertical black dashed line is required information size. The green dashed lines represent the trial sequential monitoring boundaries and the futility boundaries.

**Figure 8 jcm-08-00412-f008:**
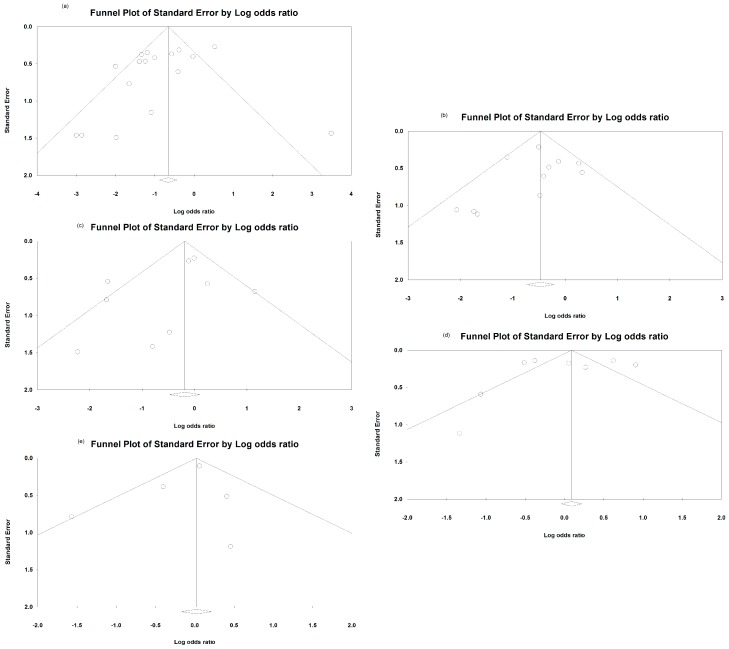
Funnel plot of the odds ratio for evaluation event: (**a**) Funnel plot of the odds ratio of catheter-related bloodstream infection. *I*^2^ value, 70.1%; *p* = 0.303; (**b**) funnel plot of the odds ratio of exit-site infection. *I*^2^ value, 28.0%; *p* = 0.010; (**c**) funnel plot of the odds ratio of catheter removal for catheter malfunction. *I*^2^ value, 55.9%; *p* = 0.208; (**d**) funnel plot of the odds ratio of thrombolytic treatments for catheter malfunction. *I*^2^ value, 88.69%; *p* = 0.946; (**e**) funnel plot of the odds ratio of all-cause mortality. *I*^2^ value, 88.6%; *p* = 0.804. Regarding odds ratio heterogeneity, the *I*^2^ value in both the overall studies included is indicated for each case. Egger’s test revealed the existence of significant publication bias regarding the overall odds ratios, *p*-value is indicated for each case.

**Table 1 jcm-08-00412-t001:** Summary of the retrieved trials investigating experimental group and control group.

Author, Year, Country, Reference	RCT	Total *N*	Treated (*N*)	Control (*N*)	QA
Buturovic et al., 1998, SI, [29]	No	30	4% CiT (20)	1666 U/mL HpR (10)	3 #
Dogra et al., 2002, AU, [30]	Yes	79	26.7 mg/mL GM + 1.04% CiT (42)	5000 U/mL HpR (37)	8 *
Betjes et al., 2004, NL, [31]	No	58	1.35% TRD +4% CiT (37)	5000 U/mL HpR (39)	3 #
Weijmer et al., 2005, NL, [14]	Yes	291	30% CiT (148)	5000 U/mL HpR (143)	8 *
Nori et al., 2006, USA, [32]	No	40	4 mg/mL GM + 3.13% CiT (41)	5000 U/mL HpR (21)	3 #
Lok et al., 2007, CA, [13]	No	250	4% CiT (129)	5000 U/mL HpR (121)	3 #
MacRae et al., 2008, CA, [12]	No	61	4% CiT (32)	5000 U/mL HpR (29)	3 #
Power et al., 2009, UK, [4]	Yes	232	46.7% CiT (132)	5000 U/mL HpR (100)	8 *
Solomon et al., 2010, UK, [33]	Yes	107	1.35% TRD + 4% CiT (53)	5000 U/mL HpR (54)	8 *
Filiopoulos et al., 2011, GR, [34]	Yes	117	1.35% TRD + 4% CiT (119)	5000 U/mL HpR (58)	8 *
Maki et al., 2011, USA, [6]	Yes	407	7.0% CiT + MMP (206)	5000 U/mL HpR (201)	8 *
Moran et al., 2012, USA, [8]	No	303	320 μg/mL GM + 4% CiT (155)	1000 U/mL HpR (148)	3 #
Chen et al., 2014, CH, [35]	Yes	72	10% NaCl (36)	3125 U/mL HpR (36)	8 *
Souweine et al., 2015, FR, [19]	Yes	1460	60% *w*/*w* EtOH (730)	0.9% NaCl (730)	8 *
Moghaddas et al., 2015, IR, [18]	Yes	87	10 mg/mL TMP/SMX + 2500 U/mL HpR (46)	2500 U/mL HpR (41)	8 *
Kanaa et al., 2015, UK, [17]	Yes	115	4% EDTA (59)	5000 U/mL HpR (56)	8 *
Zwiech et al., 2016, PL, [21]	Yes	50	4% CiT (26)	5000 U/mL HpR (24)	8 *
Chu et al., 2016, AU, [20]	Yes	100	1000 U/mL HpR (52)	5000 U/mL HpR (48)	8 *
Correa Barcellos et al., 2017, BZ, [22]	Yes	464	30% CiT (231)	5000 U/mL HpR (233)	8 *
Sofroniadou et al., 2017, GR, [23]	Yes	103	70% *w*/*w* EtOH + UFH 2000 U/mL (52)	2000 U/mL HpR (51)	8 *
Winnicki et al., 2018, Au, [24]	No	406	1.35% TRD + 4% CiT + HpR (52)	4% CiT (54)	3 #

Abbreviations: AU, Australia; Au, Austria; BZ, Brazil; CA, Canada; CH, China; CiT, citrate; EDTA, tetra-sodium ethylenediaminetetraacetic acid; EtOH, ethanol; FR, France; GM, gentamicin; GR, Greece; HpR, heparin; IR, Iran; MMP, 0.15% methylene blue + 0.15% methylparaben + 0.015% propylparaben; NaCl, sodium chloride; N, number; NL, Netherlands; PL, Poland; QA, quality assessment; RCT, randomized controlled trial; SI, Slovenia; TMP/SMX, cotrimoxazole (=trimethoprim/sulfamethoxazole); TRD, taurolidine ; UFH, unfractionated heparin; UK, United Kingdom; US, United States. #, the study was evaluated using Jadad scale. *, the study was assessed using the Newcastle-Ottawa scale.

**Table 2 jcm-08-00412-t002:** Subgroup analysis of odds ratio based on study designs, combined regimen, regimen containing antibiotic, and concentration of regimen for CRBSI.

Subgroup	Odds Ratio	95% Confidence Interval
combined regimen		
RCT	0.606	0.298–1.230
Not RCT	0.206	0.058–0.730
Not combined regimen		
RCT	0.417	0.192–0.905
Not RCT	0.289	0.083–0.365
Regimen containing antibiotic		
RCT	0.191	0.023–1.564
Not RCT	0.136	0.051–0.365
Regimen Not containing antibiotic		
RCT	0.546	0.314–0.949
Not RCT	0.342	0.191–0.614
High Concentration of major regimen		
RCT	0.644	0.155–2.671
Low Concentration of major regimen		
RCT	0.421	0.186–0.956
Not RCT	0.260	0.135–0.497

Abbreviation: RCT, randomized controlled trial.

**Table 3 jcm-08-00412-t003:** Subgroup analysis of odds ratio based on study designs, combined regimen, regimen containing antibiotic, and concentration of regimen for exit site infection.

Subgroup	Odds Ratio	95% Confidence Interval
combined regimen		
RCT	0.849	0.358–2.011
Not RCT	0.706	0.307–1.62
Not combined regimen		
RCT	0.503	0.276–0.918
Not RCT	0.620	0.113–3.389
Regimen containing antibiotic		
RCT	0.571	0.189–1.725
Not RCT	0.735	0.284–1.905
Regimen Not containing antibiotic		
RCT	0.599	0.334–1.071
Not RCT	0.650	0.246–1.722
High Concentration of major regimen		
RCT	0.631	0.214–1.862
Low Concentration of major regimen		
RCT	0.805	0.282–2.297
Not RCT	0.692	0.35–1.368

Abbreviation: RCT, randomized controlled trial.

**Table 4 jcm-08-00412-t004:** Subgroup analysis of odds ratio based on study designs, combined regimen, regimen containing antibiotic, and concentration of regimen for catheter removal due to catheter malfunction.

Subgroup	Odds Ratio	95% Confidence Interval
Combined regimen		
RCT	0.520	0.086–3.15
Not RCT	0.977	0.628–1.518
Not combined regimen		
RCT	0.434	0.068–2.786
Not RCT	1.106	0.392–3.124
Regimen containing antibiotic		
RCT	0.741	0.087–6.287
Not RCT	0.992	0.633–1.554
Regimen not containing antibiotic		
RCT	0.329	0.051–2.138
Not RCT	1.010	0.39–2.619
High concentration of major regimen		
RCT	0.896	0.029–27.554
Low concentration of major regimen		
RCT	0.479	0.051–4.537
Not RCT	0.995	0.663–1.494

Abbreviation: RCT, randomized controlled trial.

**Table 5 jcm-08-00412-t005:** Subgroup analysis of odds ratio based on study designs, combined regimen, regimen containing antibiotic, and concentration of regimen for the need of thrombolytic treatment for catheter malfunction.

Subgroup	Odds Ratio	95% Confidence Interval
Combined regimen		
RCT	2.480	1.214–5.066
Not RCT	0.620	0.382–1.004
Not combined regimen		
RCT	1.320	0.888–1.961
Not RCT	0.599	0.344–1.043
Regimen containing antibiotic		
RCT	1.969	0.944–4.107
Not RCT	0.620	0.382–1.004
Regimen not containing antibiotic		
RCT	1.385	0.893–2.149
Not RCT	0.345	0.108–1.102
High concentration of major regimen		
RCT	1.415	0.784–2.554
Low concentration of major regimen		
RCT	2.480	1.042–5.902
Not RCT	0.637	0.518–0.783

Abbreviation: RCT, randomized controlled trial.

**Table 6 jcm-08-00412-t006:** Subgroup analysis of odds ratio based on study designs, combined regimen, regimen containing antibiotic, and concentration of regimen for all-cause mortality.

Subgroup	Odds Ratio	95% Confidence Interval
Combined regimen		
RCT	0.725	0.237–2.211
Not RCT	1.579	0.154–16.18
Not combined regimen		
RCT	0.884	0.404–1.933
Regimen containing antibiotic		
RCT	1.506	0.388–5.838
Not RCT	1.579	0.154–16.18
Regimen not containing antibiotic		
RCT	0.723	0.367 – 1.425
High concentration of major regimen		
RCT	0.669	0.054–8.324
Low Concentration of major regimen		
RCT	0.615	0.09–4.22
Not RCT	1.579	0.154–16.18

Abbreviation: RCT, randomized controlled trial.

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
