# Peer review of "Re-Evaluating the Protective Effect of Hemodialysis Catheter Locking Solutions in Hemodialysis Patients"

_jcm, 2019, doi:10.3390/jcm8030412_

Reviewer 1 Report

The paper by Chen et al provides an important overview of applying locking solutions in hemodialysis catheters in order to prevent blood-stream infections, site infections, thrombosis of the catheter (with or without need to remove it) and all-cause mortality.

The methods are well described and well executed. There are some mistakes with numbers (see below). The findings largely confirm earlier meta-analyses and therefore not novel. Nevertheless the findings are important for the hemodialysis community.

Major comments:

Point (1). Figure 1: 6118 records are identified and then 458 are identified. The exclusion of the large number of articles is not clear. In the text (line 133-137) it is mentioned that 6118 were potentially relevant and 350 were irrelevant by reading the abstract. Next suddenly 108 articles are selected but it is unclear why. Please elaborate.
Next in figure 1 is mentioned that 108 articles are selected of which 95 are excluded leading to 13 articles, but finally 21 are analyzed. How did the authors get tot this number?

Point (2). Section 3.4 (line 239): the overall OR of CRBSI was 0.439 (95% CI 0,290 - 1,668; p<0.001). This OR cannot be significant if the upper CI level is above 1. So either the OR is incorrect or the upper CI is incorrect.

Point (3): discussion (line 315). In your discussion you mention that similar to other studies, your meta-analysis supports the use of citrate-based to reduce CRBSI. Of course this is similar because these studies were included in your meta-analysis and had a lot of weight in the analysis. So this statement seems void. Please elaborate.

Point (4). discussion (line 325): similar to point (3) you state that other studies support your finding about a lower incidence of ESI but again these studies were included in the meta-analysis. Please adjust this sentence.

Point (5): discussion (line 375-377). Although the findings of your meta-analysis quite strongly support the use of citrate-based locking solution you state that you cannot provide suggestions to improve clinical care because of heterogeneity. I think you can give suggestions which are likely to improve clinical care, as stated in other sections in the discussion.

Minor comments:

Point (1): Section 2.1 lines 79-97: it seems the English in this section was not corrected because multiple sentences are incorrect in terms of grammar. Please revise.

Point (2): section 2.4 line 116: "odds ratios (ORs) of PI" --> what's PI?

Point (3): section 2.4 line 118: "Or" should be "OR"

Point (4): section 3.5 line 277 (title): "or" should be "OR"

Point (5): discussion line 299: the sentence states you included a total number of participants of 6118 but in figure 1 it says you checked 6118 articles and in the results section (line 141) you state a total number of particpants of 4832. Please correct.

Point (6): discussion line 296-299: this sentence is incomprehensible, please rephrase.

Point (7): discussion line 339, "met" should be "meet"

Author Response

Author's Response to Reviewer One

Title: Re-evaluating the protective effect of hemodialysis catheter locking solutions in hemodialysis patients (ID jcm-468085)

Authors: Chen CH, et al.

Version: 2       

Date: March 12, 2019
Dear Reviewer One,

The authors thank you for your constructive comments. It is our pleasure to submit a revised manuscript that has been amended in light of your suggestions. We appreciate your comments, and we respond as follows:

# Point 1

The paper by Chen et al provides an important overview of applying locking solutions in hemodialysis catheters in order to prevent blood-stream infections, site infections, thrombosis of the catheter (with or without need to remove it) and all-cause mortality. The methods are well described and well executed. There are some mistakes with numbers (see below). The findings largely confirm earlier meta-analyses and therefore not novel. Nevertheless the findings are important for the hemodialysis community.

Response 1:

Authors thank your positive comment. Our findings are similar to previous meta-analyses reports. Our strength of our study is the novelty using both comprehensive meta-analysis and trial sequential analyses is to evaluate protective effect of hemodialysis catheter locking solutions in hemodialysis patients. In addition, the significance and importance of our study are robust at follows:

·        Primary outcome :effect of infections prevention

o   Eligible studies were included, with a total of 4,832 patients and 318,769 days of catheter use. Seventeen studies (1,731 patients; 217,128 catheter days) reported on catheter-related bloodstream infections (CRBSIs). The incidence of CRBSI was significantly lower in the treated group than in the controls [Odd ratio (OR), 0.424; 95% CI, 0.267–0.673].

o   Eleven randomized controlled trials (RCTs; 2,425 patients, 231,086 catheter days) reported on exit-site infections (ESIs). The incidence of ESI was significantly lower in the treated group than in the controls (OR, 0.627; 95% CI, 0.441–0.893).

·       Secondary outcome :effect of catheter protection

o   Nine studies (1,826 patients; 205,163 catheter days) reported the need to remove catheters due to catheter malfunction, with no between-group differences (OR, 0.696; 95% CI, 0.397–1.223).

o   The incidence of thrombolytic treatment for catheter malfunction was not significantly lower in the treated group, as per the random-effects model (OR, 1.105; 95% CI, 0.655–1.573.

·       All-cause mortality: No difference in mortality rates was found between the two groups (OR, 0.909; 95% CI, 0.580–1.423).

o   The lack of statistical significance in the latter comparisons may be attributed to the heterogeneity of the included trials and inadequate sample size.

# Point 2

Major Point (1). Figure 1: 6118 records are identified and then 458 are identified. The exclusion of the large number of articles is not clear. In the text (line 133-137) it is mentioned that 6118 were potentially relevant and 350 were irrelevant by reading the abstract. Next suddenly 108 articles are selected but it is unclear why. Please elaborate.

Next in figure 1 is mentioned that 108 articles are selected of which 95 are excluded leading to 13 articles, but finally 21 are analyzed. How did the authors get tot this number?

Response 2:  

Authors thank your comments. Authors used the search strategies listed at Appendix 1, and 458 records were identified. Authors extracted them by selecting only clinical trials enrolled and screening by title and abstract, then 100 full-text articles were assessed for eligibility (Appendix 1). Author had revised the content and Figure 1. Please see Line 133-135 and Figure 1.

# Point 3

Major Point (2). Section 3.4 (line 239): the overall OR of CRBSI was 0.439 (95% CI 0,290 - 1,668; p<0.001). This OR cannot be significant if the upper CI level is above 1. So either the OR is incorrect or the upper CI is incorrect.

Response 3:

Authors thank your comments. In trial sequential analysis between the treated and control groups, the overall OR of CRBSI was 0.439 (95% CI, 0.290–0.668; p < 0.001). We had corrected the type-error. Please see Line 247.

# Point 4

Major Point (3): discussion (line 315). In your discussion you mention that similar to other studies, your meta-analysis supports the use of citrate-based to reduce CRBSI. Of course this is similar because these studies were included in your meta-analysis and had a lot of weight in the analysis. So this statement seems void. Please elaborate.

Response 4:

Authors thank your comment. The current study provided not only results of comprehensive meta-analysis but also results of trial sequential analyses for evaluating protective effect of hemodialysis catheter locking solutions in hemodialysis patients.

Although finding no significant alteration in all-cause mortality between the two groups, we disclosed that the lack of statistical significance may not only be due to the heterogeneity and underlying variance in the outcomes of each regimen, but also to inadequate required sample sizes revealed by trial sequential analysis. Previous reports do not mention such kind required information size estimation.  We thought that our analysis is more comprehensive and robust.

# Point 5

Major Point (4). discussion (line 325): similar to point (3) you state that other studies support your finding about a lower incidence of ESI but again these studies were included in the meta-analysis. Please adjust this sentence.

Response 5:

Authors thank your positive comment. In current study, a total of 11 RCTs (2,425 patients; 231,086 catheter days) described ESI. Weijmer MC et al's study enrolled 291 patients and Souweine B et al.'s enrolled 109 patients. They are 16.5 % (400/2425) in total. Authors thought that the results of the enrolled studies (Weijmer MC et al's study and Souweine B et al.'s study) are not must-be the same with current study. Authors thought that the result is very robust for the finding about a lower incidence of ESI .

# Point 6

Major Point (5): discussion (line 375-377). Although the findings of your meta-analysis quite strongly support the use of citrate-based locking solution you state that you cannot provide suggestions to improve clinical care because of heterogeneity. I think you can give suggestions which are likely to improve clinical care, as stated in other sections in the discussion.

Response 6:

Authors thank your positive comment. Authors had revised the sentence. Please see Line 396-397.

# Point 7

Minor Point (1): Section 2.1 lines 79-97: it seems the English in this section was not corrected because multiple sentences are incorrect in terms of grammar. Please revise.

Response 7:

Authors thank your positive comment. English editing was re-checked by a professional English editing service, Editage (code: CHHEN_18_2).

# Point 8

Minor Point (2): section 2.4 line 116: "odds ratios (ORs) of PI" --> what's PI?

Response 8:

Authors thank your positive comment. The abbreviation is “primary and secondary outcomes”. Authors had revised the sentence. Please see Line 116-117.

# Point 9

Minor Point (3): section 2.4 line 118: "Or" should be "OR"

Response 9:

Authors thank your positive comment. Authors had corrected the “OR”. Please see Line 118.

# Point 10

Minor Point (4): section 3.5 line 277 (title): "or" should be "OR"

Response 10:

Authors thank your positive comment. Authors had corrected the “OR”. Please see Line 298.

# Point 11

Minor Point (5): discussion line 299: the sentence states you included a total number of participants of 6118 but in figure 1 it says you checked 6118 articles and in the results section (line 141) you state a total number of particpants of 4832. Please correct.

Response 11:

Authors thank your positive comment. Please see response to major point (1). Author had revised the content and Figure 1. Please see Line 133-135 and Figure 1. And, the total number of particpants of 4832 is correct.

# Point 12

Minor Point (6): discussion line 296-299: this sentence is incomprehensible, please rephrase.

Response 12:

Authors thank your positive comment. Authors had rewritten the sentence. Please see Line 317-319.

# Point 13

Minor Point (7): discussion line 339, "met" should be "meet"

Response 13:

Authors thank your positive comment. Authors had corrected the spelling. Please see Line 360.

The requested changes have been made. We hope that Journal of Clinical Medicine will now find our manuscript acceptable for publication.

Thank you for your consideration.

Appendix 1

Search Details

Query Translation:(("catheter-related infections"[MeSH Terms] OR ("catheter-related"[All Fields] AND "infections"[All Fields]) OR "catheter-related infections"[All Fields] OR ("catheter"[All Fields] AND "related"[All Fields] AND "infections"[All Fields]) OR "catheter related infections"[All Fields]) AND ("catheter-related infections"[MeSH Terms] OR ("catheter-related"[All Fields] AND "infections"[All Fields]) OR "catheter-related infections"[All Fields] OR ("catheter"[All Fields] AND "related"[All Fields] AND "infection"[All Fields]) OR "catheter related infection"[All Fields]) AND ("catheter-related infections"[MeSH Terms] OR ("catheter-related"[All Fields] AND "infections"[All Fields]) OR "catheter-related infections"[All Fields] OR ("infection"[All Fields] AND "catheter"[All Fields] AND "related"[All Fields])) AND ("catheter-related infections"[MeSH Terms] OR ("catheter-related"[All Fields] AND "infections"[All Fields]) OR "catheter-related infections"[All Fields] OR ("infections"[All Fields] AND "catheter"[All Fields] AND "related"[All Fields])) AND ("catheter-related infections"[MeSH Terms] OR ("catheter-related"[All Fields] AND "infections"[All Fields]) OR "catheter-related infections"[All Fields] OR ("catheter"[All Fields] AND "associated"[All Fields] AND "infections"[All Fields]) OR "catheter associated infections"[All Fields]) AND ("catheter-related infections"[MeSH Terms] OR ("catheter-related"[All Fields] AND "infections"[All Fields]) OR "catheter-related infections"[All Fields] OR ("catheter"[All Fields] AND "associated"[All Fields] AND "infections"[All Fields]) OR "catheter associated infections"[All Fields]) AND ("catheter-related infections"[MeSH Terms] OR ("catheter-related"[All Fields] AND "infections"[All Fields]) OR "catheter-related infections"[All Fields] OR ("catheter"[All Fields] AND "associated"[All Fields] AND "infection"[All Fields]) OR "catheter associated infection"[All Fields]) AND ("catheter-related infections"[MeSH Terms] OR ("catheter-related"[All Fields] AND "infections"[All Fields]) OR "catheter-related infections"[All Fields] OR ("infection"[All Fields] AND "catheter"[All Fields] AND "associated"[All Fields]) OR "infection, catheter associated"[All Fields]) AND ("catheter-related infections"[MeSH Terms] OR ("catheter-related"[All Fields] AND "infections"[All Fields]) OR "catheter-related infections"[All Fields] OR ("infections"[All Fields] AND "catheter"[All Fields] AND "associated"[All Fields]))) AND (("anti-infective agents, local"[Pharmacological Action] OR "anti-infective agents, local"[MeSH Terms] OR ("anti-infective"[All Fields] AND "agents"[All Fields] AND "local"[All Fields]) OR "local anti-infective agents"[All Fields] OR ("anti"[All Fields] AND "infective"[All Fields] AND "agents"[All Fields] AND "local"[All Fields]) OR "anti infective agents, local"[All Fields]) AND "anti-infective agents, local"[Pharmacological Action] OR "anti-infective agents, local"[MeSH Terms] OR ("anti-infective"[All Fields] AND "agents"[All Fields] AND "local"[All Fields]) OR "local anti-infective agents"[All Fields] OR ("anti"[All Fields] AND "infective"[All Fields] AND "agents"[All Fields] AND "local"[All Fields]) OR "anti infective agents, local"[All Fields] AND ("disinfectants"[Pharmacological Action] OR "disinfectants"[MeSH Terms] OR "disinfectants"[All Fields]) AND "disinfectants"[Pharmacological Action] OR "disinfectants"[MeSH Terms] OR "disinfectants"[All Fields] AND ("anti-infective agents, local"[Pharmacological Action] OR "anti-infective agents, local"[MeSH Terms] OR ("anti-infective"[All Fields] AND "agents"[All Fields] AND "local"[All Fields]) OR "local anti-infective agents"[All Fields] OR "antiseptic"[All Fields]) AND "anti- infective agents, local"[Pharmacological Action] OR "anti-infective agents, local"[MeSH Terms] OR ("anti-infective"[All Fields] AND "agents"[All Fields] AND "local"[All Fields]) OR "local anti-infective agents"[All Fields] OR "antiseptic"[All Fields])

Search URL

Result: 458

Translations: Catheter Associated Infections"catheter-related infections"[MeSH Terms] OR ("catheter-related"[All Fields] AND "infections"[All Fields]) OR "catheter-related infections"[All Fields] OR ("catheter"[All Fields] AND "associated"[All Fields] AND "infections"[All Fields]) OR "catheter associated infections"[All Fields] Catheter Related Infections    "catheter-related infections"[MeSH Terms] OR ("catheter-related"[All Fields] AND "infections"[All Fields]) OR "catheter-related infections"[All Fields] OR ("catheter"[All Fields] AND "related"[All Fields] AND "infections"[All Fields]) OR "catheter related infections"[All Fields] Catheter-Associated Infection "catheter-related infections"[MeSH Terms] OR ("catheter-related"[All Fields] AND "infections"[All Fields]) OR "catheter-related infections"[All Fields] OR ("catheter"[All Fields] AND "associated"[All Fields] AND "infection"[All Fields]) OR "catheter associated infection"[All Fields] Catheter-Associated Infections          "catheter-related infections"[MeSH Terms] OR ("catheter-related"[All Fields] AND "infections"[All Fields]) OR "catheter-related infections"[All Fields] OR ("catheter"[All Fields] AND "associated"[All Fields] AND "infections"[All Fields]) OR "catheter associated infections"[All Fields] Catheter-Related Infection            "catheter-related infections"[MeSH Terms] OR ("catheter-related"[All Fields] AND "infections"[All Fields]) OR "catheter-related infections"[All Fields] OR ("catheter"[All Fields] AND "related"[All Fields] AND "infection"[All Fields]) OR "catheter related infection"[All Fields] Infection, Catheter-Associated "catheter-related infections"[MeSH Terms] OR ("catheter-related"[All Fields] AND "infections"[All Fields]) OR "catheter-related infections"[All Fields] OR ("infection"[All Fields] AND "catheter"[All Fields] AND "associated"[All Fields]) OR "infection, catheter associated"[All Fields] Infection, Catheter-Related "catheter-related infections"[MeSH Terms] OR ("catheter-related"[All Fields] AND "infections"[All Fields]) OR "catheter-related infections"[All Fields] OR ("infection"[All Fields] AND "catheter"[All Fields] AND "related"[All Fields]) Infections, Catheter-Associated "catheter-related infections"[MeSH Terms] OR ("catheter-related"[All Fields] AND "infections"[All Fields]) OR "catheter-related infections"[All Fields] OR ("infections"[All Fields] AND "catheter"[All Fields] AND "associated"[All Fields]) Infections, Catheter-Related "catheter-related infections"[MeSH Terms] OR ("catheter-related"[All Fields] AND "infections"[All Fields]) OR "catheter-related infections"[All Fields] OR ("infections"[All Fields] AND "catheter"[All Fields] AND "related"[All Fields]) Anti-Infective Agents, Local "anti-infective agents, local"[Pharmacological Action] OR "anti-infective agents, local"[MeSH Terms] OR ("anti-infective"[All Fields] AND "agents"[All Fields] AND "local"[All Fields]) OR "local anti-infective agents"[All Fields] OR ("anti"[All Fields] AND "infective"[All Fields] AND "agents"[All Fields] AND "local"[All Fields]) OR "anti infective agents, local"[All Fields]Disinfectants  disinfectants"[Pharmacological Action] OR "disinfectants"[MeSH Terms] OR "disinfectants"[All Fields] antiseptic "anti-infective agents, local"[Pharmacological Action] OR "anti-infective agents, local"[MeSH Terms] OR ("anti-infective"[All Fields] AND "agents"[All Fields] AND "local"[All Fields]) OR "local anti-infective agents"[All Fields] OR "antiseptic"[All Fields]

Database: PubMed

User query:(Catheter Related Infections Catheter-Related Infection Infection, Catheter-Related Infections, Catheter-Related Catheter-Associated Infections Catheter Associated Infections Catheter-Associated Infection Infection, Catheter-Associated Infections, Catheter-Associated) AND (Anti-Infective Agents, Local "anti-infective agents, local"[Pharmacological Action] OR "anti-infective agents, local"[MeSH Terms] OR ("anti-infective"[All Fields] AND "agents"[All Fields] AND "local"[All Fields]) OR "local anti-infective agents"[All Fields] OR ("anti"[All Fields] AND "infective"[All Fields] AND "agents"[All Fields] AND "local"[All Fields]) OR "anti infective agents, local"[All Fields] Disinfectants "disinfectants"[Pharmacological Action] OR "disinfectants"[MeSH Terms] OR "disinfectants"[All Fields] antiseptic "anti- infective agents, local"[Pharmacological Action] OR "anti-infective agents, local"[MeSH Terms] OR ("anti-infective"[All Fields] AND "agents"[All Fields] AND "local"[All Fields]) OR "local anti-infective agents"[All Fields] OR "antiseptic"[All Fields])

Appendix 2

Search Details

Query Translation: (("catheter-related infections"[MeSH Terms] OR ("catheter-related"[All Fields] AND "infections"[All Fields]) OR "catheter-related infections"[All Fields] OR ("catheter"[All Fields] AND "related"[All Fields] AND "infections"[All Fields]) OR "catheter related infections"[All Fields]) AND ("catheter-related infections"[MeSH Terms] OR ("catheter-related"[All Fields] AND "infections"[All Fields]) OR "catheter-related infections"[All Fields] OR ("catheter"[All Fields] AND "related"[All Fields] AND "infection"[All Fields]) OR "catheter related infection"[All Fields]) AND ("catheter-related infections"[MeSH Terms] OR ("catheter-related"[All Fields] AND "infections"[All Fields]) OR "catheter-related infections"[All Fields] OR ("infection"[All Fields] AND "catheter"[All Fields] AND "related"[All Fields])) AND ("catheter-related infections"[MeSH Terms] OR ("catheter-related"[All Fields] AND "infections"[All Fields]) OR "catheter-related infections"[All Fields] OR ("infections"[All Fields] AND "catheter"[All Fields] AND "related"[All Fields])) AND ("catheter-related infections"[MeSH Terms] OR ("catheter-related"[All Fields] AND "infections"[All Fields]) OR "catheter-related infections"[All Fields] OR ("catheter"[All Fields] AND "associated"[All Fields] AND "infections"[All Fields]) OR "catheter associated infections"[All Fields]) AND ("catheter-related infections"[MeSH Terms] OR ("catheter-related"[All Fields] AND "infections"[All Fields]) OR "catheter-related infections"[All Fields] OR ("catheter"[All Fields] AND "associated"[All Fields] AND "infections"[All Fields]) OR "catheter associated infections"[All Fields]) AND ("catheter-related infections"[MeSH Terms] OR ("catheter-related"[All Fields] AND "infections"[All Fields]) OR "catheter-related infections"[All Fields] OR ("catheter"[All Fields] AND "associated"[All Fields] AND "infection"[All Fields]) OR "catheter associated infection"[All Fields]) AND ("catheter-related infections"[MeSH Terms] OR ("catheter-related"[All Fields] AND "infections"[All Fields]) OR "catheter-related infections"[All Fields] OR ("infection"[All Fields] AND "catheter"[All Fields] AND "associated"[All Fields]) OR "infection, catheter associated"[All Fields]) AND ("catheter-related infections"[MeSH Terms] OR ("catheter-related"[All Fields] AND "infections"[All Fields]) OR "catheter-related infections"[All Fields] OR ("infections"[All Fields] AND "catheter"[All Fields] AND "associated"[All Fields]))) AND (("anti-infective agents, local"[Pharmacological Action] OR "anti-infective agents, local"[MeSH Terms] OR ("anti-infective"[All Fields] AND "agents"[All Fields] AND "local"[All Fields]) OR "local anti-infective agents"[All Fields] OR ("anti"[All Fields] AND "infective"[All Fields] AND "agents"[All Fields] AND "local"[All Fields]) OR "anti infective agents, local"[All Fields]) AND "anti-infective agents, local"[Pharmacological Action] OR "anti-infective agents, local"[MeSH Terms] OR ("anti-infective"[All Fields] AND "agents"[All Fields] AND "local"[All Fields]) OR "local anti-infective agents"[All Fields] OR ("anti"[All Fields] AND "infective"[All Fields] AND "agents"[All Fields] AND "local"[All Fields]) OR "anti infective agents, local"[All Fields] AND ("disinfectants"[Pharmacological Action] OR "disinfectants"[MeSH Terms] OR "disinfectants"[All Fields]) AND "disinfectants"[Pharmacological Action] OR "disinfectants"[MeSH Terms] OR "disinfectants"[All Fields] AND ("anti-infective agents, local"[Pharmacological Action] OR "anti-infective agents, local"[MeSH Terms] OR ("anti-infective"[All Fields] AND "agents"[All Fields] AND "local"[All Fields]) OR "local anti-infective agents"[All Fields] OR "antiseptic"[All Fields]) AND "anti-infective agents, local"[Pharmacological Action] OR "anti-infective agents, local"[MeSH Terms] OR ("anti-infective"[All Fields] AND "agents"[All Fields] AND "local"[All Fields]) OR "local anti-infective agents"[All Fields] OR "antiseptic"[All Fields]) AND Clinical Trial[ptyp]

Search URL

Result: 108

Translations: Catheter Associated Infections "catheter-related infections"[MeSH Terms] OR ("catheter-related"[All Fields] AND "infections"[All Fields]) OR "catheter-related infections"[All Fields] OR ("catheter"[All Fields] AND "associated"[All Fields] AND "infections"[All Fields]) OR "catheter associated infections"[All Fields] Catheter Related Infections "catheter-related infections"[MeSH Terms] OR ("catheter-related"[All Fields] AND "infections"[All Fields]) OR "catheter-related infections"[All Fields] OR ("catheter"[All Fields] AND "related"[All Fields] AND "infections"[All Fields]) OR "catheter related infections"[All Fields] Catheter-Associated Infection "catheter-related infections"[MeSH Terms] OR ("catheter-related"[All Fields] AND "infections"[All Fields]) OR "catheter-related infections"[All Fields] OR ("catheter"[All Fields] AND "associated"[All Fields] AND "infection"[All Fields]) OR "catheter associated infection"[All Fields] Catheter-Associated Infections "catheter-related infections"[MeSH Terms] OR ("catheter-related"[All Fields] AND "infections"[All Fields]) OR "catheter-related infections"[All Fields] OR ("catheter"[All Fields] AND "associated"[All Fields] AND "infections"[All Fields]) OR "catheter associated infections"[All Fields] Catheter-Related Infection "catheter-related infections"[MeSH Terms] OR ("catheter-related"[All Fields] AND "infections"[All Fields]) OR "catheter-related infections"[All Fields] OR ("catheter"[All Fields] AND "related"[All Fields] AND "infection"[All Fields]) OR "catheter related infection"[All Fields] Infection, Catheter-Associated "catheter-related infections"[MeSH Terms] OR ("catheter-related"[All Fields] AND "infections"[All Fields]) OR "catheter-related infections"[All Fields] OR ("infection"[All Fields] AND "catheter"[All Fields] AND "associated"[All Fields]) OR "infection, catheter associated"[All Fields] Infection, Catheter-Related "catheter-related infections"[MeSH Terms] OR ("catheter-related"[All Fields] AND "infections"[All Fields]) OR "catheter-related infections"[All Fields] OR ("infection"[All Fields] AND "catheter"[All Fields] AND "related"[All Fields]) Infections, Catheter-Associated "catheter-related infections"[MeSH Terms] OR ("catheter-related"[All Fields] AND "infections"[All Fields]) OR "catheter-related infections"[All Fields] OR ("infections"[All Fields] AND "catheter"[All Fields] AND "associated"[All Fields]) Infections, Catheter-Related "catheter-related infections"[MeSH Terms] OR ("catheter-related"[All Fields] AND "infections"[All Fields]) OR "catheter-related infections"[All Fields] OR ("infections"[All Fields] AND "catheter"[All Fields] AND "related"[All Fields]) Anti-Infective Agents, Local "anti-infective agents, local"[Pharmacological Action] OR "anti-infective agents, local"[MeSH Terms] OR ("anti-infective"[All Fields] AND "agents"[All Fields] AND "local"[All Fields]) OR "local anti-infective agents"[All Fields] OR ("anti"[All Fields] AND "infective"[All Fields] AND "agents"[All Fields] AND "local"[All Fields]) OR "anti infective agents, local"[All Fields] Disinfectants "disinfectants"[Pharmacological Action] OR "disinfectants"[MeSH Terms] OR "disinfectants"[All Fields] antiseptic "anti-infective agents, local"[Pharmacological Action] OR "anti-infective agents, local"[MeSH Terms] OR ("anti-infective"[All Fields] AND "agents"[All Fields] AND "local"[All Fields]) OR "local anti-infective agents"[All Fields] OR "antiseptic"[All Fields] 

Database: PubMed

User query: (Catheter Related Infections,Catheter-Related Infection,Infection, Catheter-Related,Infections, Catheter-Related,Catheter-Associated Infections,Catheter Associated Infections,Catheter-Associated Infection, Infection, Catheter-Associated,Infections, Catheter-Associated) AND (Anti-Infective Agents, Local "anti-infective agents, local"[Pharmacological Action] OR "anti-infective agents, local"[MeSH Terms] OR ("anti-infective"[All Fields] AND "agents"[All Fields] AND "local"[All Fields]) OR "local anti-infective agents"[All Fields] OR ("anti"[All Fields] AND "infective"[All Fields] AND "agents"[All Fields] AND "local"[All Fields]) OR "anti infective agents, local"[All Fields] Disinfectants "disinfectants"[Pharmacological Action] OR "disinfectants"[MeSH Terms] OR "disinfectants"[All Fields] antiseptic "anti-infective agents, local"[Pharmacological Action] OR "anti-infective agents, local"[MeSH Terms] OR ("anti-infective"[All Fields] AND "agents"[All Fields] AND "local"[All Fields]) OR "local anti-infective agents"[All Fields] OR "antiseptic"[All Fields]) AND (Clinical Trial[ptyp])

Reviewer 2 Report

Chen et al. evaluated that the hemodialysis catheter locking solutions reduce the

incidence of catheter-related bloodstream infections (CRBISs) and exit-site infections (ESIs) and that no benefit was found between the treated group and the controls for hemodialysis catheters to preserve catheter function and mortality.  Because the evidence of the protective effect of hemodialysis catheter locking solutions in hemodialysis patients remains insufficient, this manuscript is interesting.  However, some serious concerns have been raised.

Major comments: 

(1)  The authors indicated that the benefit of routine locking solutions for hemodialysis

catheters for preserving catheter function and mortality rates cannot be concluded by this meta-analysis because of heterogeneity and inadequate sample size.  If so, the authors should delete the description, and indicate the effectiveness after the well-conducted observational studies and randomized controlled trials are conclusively evaluated about the catheter function and all-cause mortality.

 (2)  The authors indicated the effectiveness for exit-site infection by citrate alone or the

mixtures of citrate and other antimicrobials compared with heparin locks.  I do not know why drugs to fill the catheter have an effect on exit-site infection.  The authors should indicate the reasons.

 (3)  Even though all of the subgroup analyses by combined regimen, regimen containing

antibiotic, and concentration of regimen for exit site infection were not effective, the incidence of exit site infection was significantly lower in the treated group compared with the control group.  I do not know the reason of the discrepancy of the results.  Therefore, the authors should explain it.

Minor comments:

(1)  In page 9, lines 210-211, the authors indicated that “The incidence of thrombolytic

treatment was significantly lower in the treated group compared to the control group”.  However, according to OR, 1.105; 95% CI, 0.655-1.573; p=0.946, the incidence of thrombolytic treatment was not significantly lower in the treated group compared to the control group.  Therefore, the authors should revise the sentence adequately.

 (2)  Because the letters of Figure 7 are small and the resolution of Figure 7 is not good,

it is difficult to identify Figure 7.  Therefore, the authors should revise Figure 7 adequately.

Author Response

Author's Response to Reviewer Two

Title: Re-evaluating the protective effect of hemodialysis catheter locking solutions in hemodialysis patients (ID jcm-468085)

Authors: Chen CH, et al.

Version: 2       

Date: March 12, 2019

Dear Reviewer Two,

The authors thank you for your constructive comments. It is our pleasure to submit a revised manuscript that has been amended in light of your suggestions. We appreciate your comments, and we respond as follows:

# Point 1

Major point (1)  The authors indicated that the benefit of routine locking solutions for hemodialysis catheters for preserving catheter function and mortality rates cannot be concluded by this meta-analysis because of heterogeneity and inadequate sample size. If so, the authors should delete the description, and indicate the effectiveness after the well-conducted observational studies and randomized controlled trials are conclusively evaluated about the catheter function and all-cause mortality.

Response 1:

Authors thank your positive comment. Authors thought that preserving catheter function is a very important issue for the hemodialysis catheter locking solutions in hemodialysis patients. Recent many efforts have tried to minimize the use of catheters for hemodialysis as well as have disclosed some protective strategies to prevent CRBSI and catheter malfunction in hemodialysis patients. Authors thought that both prevent CRBSI and preserve catheter function are equally critical. Please allow us to keep the analysis for the preserving catheter function.

Authors want to emphasize that our strength of our study is the novelty using both comprehensive meta-analysis and trial sequential analyses is to evaluate protective effect of hemodialysis catheter locking solutions in hemodialysis patients. In addition, the significance and importance of our study are robust at follows:

·        Primary outcome :effect of infections prevention

o   Eligible studies were included, with a total of 4,832 patients and 318,769 days of catheter use. Seventeen studies (1,731 patients; 217,128 catheter days) reported on catheter-related bloodstream infections (CRBSIs). The incidence of CRBSI was significantly lower in the treated group than in the controls [Odd ratio (OR), 0.424; 95% CI, 0.267–0.673].

o   Eleven randomized controlled trials (RCTs; 2,425 patients, 231,086 catheter days) reported on exit-site infections (ESIs). The incidence of ESI was significantly lower in the treated group than in the controls (OR, 0.627; 95% CI, 0.441–0.893).

·       Secondary outcome :effect of catheter protection

o   Nine studies (1,826 patients; 205,163 catheter days) reported the need to remove catheters due to catheter malfunction, with no between-group differences (OR, 0.696; 95% CI, 0.397–1.223).

o   The incidence of thrombolytic treatment for catheter malfunction was not significantly lower in the treated group, as per the random-effects model (OR, 1.105; 95% CI, 0.655–1.573.

·       All-cause mortality: No difference in mortality rates was found between the two groups (OR, 0.909; 95% CI, 0.580–1.423).

o   The lack of statistical significance in the latter comparisons may be attributed to the heterogeneity of the included trials and inadequate sample size.

# Point 2

Major point (2)  The authors indicated the effectiveness for exit-site infection by citrate alone or the mixtures of citrate and other antimicrobials compared with heparin locks.  I do not know why drugs to fill the catheter have an effect on exit-site infection.  The authors should indicate the reasons.

Response 2:

Authors thank your positive comment. Author thought lock solution could prevent catheter-related infection as well as exit-site infection.

# Point 3

Major point (3)  Even though all of the subgroup analyses by combined regimen, regimen containing antibiotic, and concentration of regimen for exit site infection were not effective, the incidence of exit site infection was significantly lower in the treated group compared with the control group.  I do not know the reason of the discrepancy of the results.  Therefore, the authors should explain it.

Response 3:

Authors thank your positive comment. The reason of the discrepancy of the results is bias. Whenever a subgroup analysis is performed, the randomization of patient characteristics between the treatment group and the control group is no longer necessarily maintained. Consider a subgroup analysis according to combined regimen, regimen containing antibiotic, or concentration of regimen. The randomization process should ensure, if the sample is large enough, that the treatment and control groups are balanced according to three arms. But randomization does not ensure that the two groups are balanced within those strata.

# Point 4

Minor point (1)  In page 9, lines 210-211, the authors indicated that “The incidence of thrombolytic treatment was significantly lower in the treated group compared to the control group”.  However, according to OR, 1.105; 95% CI, 0.655-1.573; p=0.946, the incidence of thrombolytic treatment was not significantly lower in the treated group compared to the control group.  Therefore, the authors should revise the sentence adequately.

Response 4:

Authors thank your positive comment. Authors had revised the sentence. Please see Line 218.

# Point 5

Minor point (2)  Because the letters of Figure 7 are small and the resolution of Figure 7 is not good, it is difficult to identify Figure 7.  Therefore, the authors should revise Figure 7 adequately.

Response 5:

Authors thank your positive comment. Author had revised the Figure 7. Please see Figure 7.

The requested changes have been made. We hope that Journal of Clinical Medicine will now find our manuscript acceptable for publication.

Thank you for your consideration.

Round  2

Reviewer 1 Report

The authors have improved the manuscript. However, the first sentence of the discussion is still incomprehensible and readers of the journal will not understand this sentence due to grammatical errors. 

The last sentence of the discussion makes absolutely no sence whatsoever. I do agree that despite the heterogeneity the authors can make a positive statement on the use of locking solutions. Stating that they "could" has no added value. I don't think the authors had the manuscript checked again with the English editing department. 

My suggestion: remove the last sentence of the discussion. 

Author Response

p.p1 {margin: 0.0px 0.0px 0.0px 0.0px; text-align: justify; font: 12.0px 'Times New Roman'; color: #000000; -webkit-text-stroke: #000000} p.p2 {margin: 0.0px 0.0px 0.0px 0.0px; text-align: justify; font: 12.0px 'Times New Roman'; color: #000000; -webkit-text-stroke: #000000; min-height: 15.0px} p.p3 {margin: 0.0px 0.0px 0.0px 0.0px; text-align: justify; text-indent: 24.0px; font: 12.0px 'Times New Roman'; color: #000000; -webkit-text-stroke: #000000} p.p4 {margin: 0.0px 0.0px 0.0px 0.0px; text-align: justify; font: 12.0px 'Times New Roman'; color: #fb0007; -webkit-text-stroke: #fb0007} p.p5 {margin: 0.0px 0.0px 0.0px 0.0px; text-align: justify; text-indent: 24.0px; font: 12.0px 'Times New Roman'; color: #fb0007; -webkit-text-stroke: #fb0007} p.p6 {margin: 0.0px 0.0px 0.0px 0.0px; text-align: justify; text-indent: 24.0px; font: 12.0px 'Times New Roman'; color: #000000; -webkit-text-stroke: #000000; min-height: 15.0px} p.p7 {margin: 0.0px 0.0px 0.0px 0.0px; text-align: center; text-indent: 24.0px; font: 12.0px 'Times New Roman'; color: #000000; -webkit-text-stroke: #000000} p.p8 {margin: 0.0px 0.0px 0.0px 0.0px; text-align: right; text-indent: 24.0px; font: 12.0px 'Times New Roman'; color: #000000; -webkit-text-stroke: #000000} li.li4 {margin: 0.0px 0.0px 0.0px 0.0px; text-align: justify; font: 12.0px 'Times New Roman'; color: #fb0007; -webkit-text-stroke: #fb0007} span.s1 {font-kerning: none} span.s2 {font: 12.0px Wingdings} span.Apple-tab-span {white-space:pre} ul.ul1 {list-style-type: disc}

Author's Response to Reviewer One

Title: Re-evaluating the protective effect of hemodialysis catheter locking solutions in hemodialysis patients (ID jcm-468085)

Authors: Chen CH, et al.

Version: 2 

Date: March 18, 2019

Dear Reviewer One, 

The authors thank you for your constructive comments. It is our pleasure to submit a revised manuscript that has been amended in light of your suggestions. Revisions highlighted with red color. We appreciate your comments, and we respond as follows:

# Point 1

The authors have improved the manuscript. However, the first sentence of the discussion is still incomprehensible and readers of the journal will not understand this sentence due to grammatical errors. The last sentence of the discussion makes absolutely no sence whatsoever. I do agree that despite the heterogeneity the authors can make a positive statement on the use of locking solutions. Stating that they "could" has no added value. I don't think the authors had the manuscript checked again with the English editing department. My suggestion: remove the last sentence of the discussion.

Response 1:

Authors thank your positive comment. English editing had been re-checked by a professional English editing service, Editage (code: CHHEN_18_2). English editing certification is attached. 

Authors had revised the first paragraph of the Discussion Section

Before : Our meta-analysis and trial sequential analysis focused on routine locking solutions for hemodialysis catheters could effectively reduce the incidence of CRBSI and ESI. And, the results for preserving catheter function, and all-cause mortality in hemodialysis patients are still non-concluded. Our current meta-analysis, based on 21 enrolled studies with a total of 6,118 participants, showed that the incidence of CRBSI and ESI significantly decreased in the treated group relative to the control group, that is less infections when using citrate or citrate mixtures versus heparin. Moreover, we found no significant difference in preserving catheter function, including in the need for catheter withdrawal or for thrombolytic treatment due to catheter malfunction, between the treated and control groups. We found no significant alteration in all-cause mortality between the two groups. The lack of statistical significance may not only be due to the heterogeneity and underlying variance in the outcomes of each regimen, but also to inadequate required sample sizes revealed by trial sequential analysis. Regular lock care with citrate is standard practice for hemodialysis patients in many healthcare institutes, but not in some countries including Taiwan. However, our updated review suggests that the benefit of routine locking solutions for preventing CRBSI and ESI in hemodialysis patients is overestimated. Additionally, it does not show a benefit in preserving catheter function in hemodialysis patients, including in the need to remove catheters or in the need for thrombolytic treatment for catheter malfunction.

After: Our meta-analysis and trial sequential analysis shows that routine locking solutions for hemodialysis catheters could effectively reduce the incidence of CRBSI and ESI. Our current meta-analysis, based on 21 selected studies with a total of 6,118 participants, showed that the incidence of CRBSI and ESI significantly decreased in the treated group relative to the control group, that is less infections when using citrate or citrate mixtures versus heparin. Moreover, we found no significant difference in preserving catheter function, including in the need for catheter withdrawal or for thrombolytic treatment due to catheter malfunction, between the treated and control groups. We found no significant alteration in all-cause mortality between the two groups. The lack of statistical significance may not only be due to the heterogeneity and underlying variance in the outcomes of each regimen, but also due to inadequate required information sizes, as revealed by the trial sequential analysis. Regular locking care with citrate is standard practice for patients undergoing hemodialysis in many healthcare institutes, but not in some countries including Taiwan. Our updated review suggests that the role of routine locking solutions in preventing CRBSI and ESI in hemodialysis patients is robust. However, it does not show a benefit in preserving catheter function in hemodialysis patients, including in the need to remove catheters or in the need for thrombolytic treatment for catheter malfunction. 

Authors had revised the last sentence of Discussion Section

Before : In spite of the heterogeneity in the comparisons and measured effects of the trials and inadequate sample size, we could provide suggestions to improve the clinical care.

After: Regardless of aforementioned limitations, we have minimized bias throughout the process by our methods of study identification, data selection, and statistical analysis, as well as in our control of publication bias. These steps should strengthen the stability and accuracy of the meta-analysis. And our findings of this meta-analysis are reliable to provide suggestions for improving the clinical care.

Authors had revised the last sentence of the discussion

Authors thank for you feedback on the readability of this manuscript, and we believe this manuscript is more clear and readable.

The requested changes have been made. We hope that Journal of Clinical Medicine will now find our manuscript acceptable for publication.

Thank you for your consideration.

Reviewer 2 Report

The authors' reply have convinced me.  I have no special comments about this revised manuscript.

Author Response

p.p1 {margin: 0.0px 0.0px 0.0px 0.0px; text-align: justify; font: 12.0px 'Times New Roman'; color: #000000; -webkit-text-stroke: #000000} p.p2 {margin: 0.0px 0.0px 0.0px 0.0px; text-align: justify; font: 12.0px 'Times New Roman'; color: #000000; -webkit-text-stroke: #000000; min-height: 15.0px} p.p3 {margin: 0.0px 0.0px 0.0px 0.0px; text-align: justify; text-indent: 24.0px; font: 12.0px 'Times New Roman'; color: #000000; -webkit-text-stroke: #000000} p.p4 {margin: 0.0px 0.0px 0.0px 0.0px; text-align: justify; font: 12.0px 'Times New Roman'; color: #fb0007; -webkit-text-stroke: #fb0007} p.p5 {margin: 0.0px 0.0px 0.0px 0.0px; text-align: justify; text-indent: 24.0px; font: 12.0px 'Times New Roman'; color: #fb0007; -webkit-text-stroke: #fb0007} p.p6 {margin: 0.0px 0.0px 0.0px 0.0px; text-align: center; text-indent: 24.0px; font: 12.0px 'Times New Roman'; color: #000000; -webkit-text-stroke: #000000} p.p7 {margin: 0.0px 0.0px 0.0px 0.0px; text-align: right; text-indent: 24.0px; font: 12.0px 'Times New Roman'; color: #000000; -webkit-text-stroke: #000000} span.s1 {font-kerning: none} span.Apple-tab-span {white-space:pre}

Author's Response to Reviewer Two

Title: Re-evaluating the protective effect of hemodialysis catheter locking solutions in hemodialysis patients (ID jcm-468085)

Authors: Chen CH, et al.

Version: 3 

Date: March 18, 2019

Dear Reviewer Two, 

The authors thank you for your constructive comments. It is our pleasure to submit a revised manuscript that has been amended in light of your suggestions. Revisions highlighted with red color. We appreciate your comments, and we respond as follows:

# Point 1

The authors' reply have convinced me.  I have no special comments about this revised manuscript.

Response 1:

Authors thank your positive comment. 

We hope that Journal of Clinical Medicine will now find our manuscript acceptable for publication.

Thank you for your consideration.

Sincerely yours,

Chang-Hua Chen, M.D.,M.Sc., Ph.D.